# Ethylene Recovery via Pebax-Based Composite Membrane: Numerical Optimization

**Nadia Hartini Suhaimi** [1,2], **Norwahyu Jusoh** [1,2,*], **Syafeeqa Syaza Rashidi** [1], **Christine Wei Mann Ch'ng** [1,2] **and Nonni Soraya Sambudi** [3]

1. Department of Chemical Engineering, Universiti Teknologi PETRONAS, Seri Iskandar 32610, Malaysia
2. CO$_2$ Research Centre (CO$_2$RES), R&D Building, Universiti Teknologi PETRONAS, Seri Iskandar 32610, Malaysia
3. Chemical Engineering Study, Universitas Pertamina, Jl. Teuku Nyak Arief, RT.7/RW.8, Simprug, Kec. Kby. Lama, Kota Jakarta Selatan, Daerah Khusus Ibukota Jakarta 12220, Indonesia
* Correspondence: norwahyu.jusoh@utp.edu.my

**Abstract:** Membrane technology, particularly polymeric membranes, is utilized in major industrial ethylene recovery owing to the very convenient and robust process. Thus, in this paper, a composite membrane (CM) comprising SAPO-34 and Pebax-1657 was employed to conduct a separation performance under two operating conditions, including temperatures and pressures, ranging from 25.0–60.0 °C and 3.5–10.0 bar, respectively. CO$_2$ permeability and CO$_2$/C$_2$H$_4$ ideal selectivity values that ranged from 105.68 to 262.86 Barrer and 1.81 to 3.52, respectively, were obtained via the experimental works. The separation of carbon dioxide (CO$_2$) from ethylene (C$_2$H$_4$) has then been optimized using response surface methodology (RSM) by adopting a central composite design (CCD) method. As a result, the ideal operational conditions were discovered at a temperature of 60.0 °C and pressure of 10.0 bar with the maximum CO$_2$ permeability of 233.62 Barrer and CO$_2$/C$_2$H$_4$ ideal selectivity of 3.22. The typical discrepancies between experimental and anticipated data for CO$_2$ permeability and CO$_2$/C$_2$H$_4$ ideal selectivity were 1.67% and 3.10%, respectively, demonstrating the models' validity. Overall, a new combination of Pebax-1657 and SAPO-34 composite membrane could inspire the latest understanding of the ethylene recovery process.

**Keywords:** CO$_2$/C$_2$H$_4$ separation; composite membrane; optimization; response surface methodology





## 1. Introduction

The worldwide ethylene (C$_2$H$_4$) market is expected to be valued at around USD 245,005 million by 2027. C$_2$H$_4$ is commonly used as a basic feedstock in the manufacture of polyethylene (PE), ethylene oxide (EO), ethylene glycol, high-density polyethylene, vinyl acetate monomers (VAM), and other chemicals. For instance, EO is a chemical precursor that has been utilized for the production of plastics, textiles, detergents, and other downstream chemicals [1]. It is produced by mixing ethylene and oxygen (O$_2$) in a catalytic reactor; however, this process also produces CO$_2$ and argon (Ar), which must be purged [2]. Unfortunately, a significant quantity of ethylene is lost during purging in order to prevent impurities from entering the catalytic reactor from the gas mixture. As a result, C$_2$H$_4$ recovery and recycling have become significant challenges in the petrochemical business particularly in EO and VAM plants.

Few conventional technologies are employed for removing CO$_2$ from C$_2$H$_4$, including absorption, adsorption, cryogenic distillation and membrane separation technologies [3]. The challenges in using cryogenic distillation are the high cost as well as energy inefficiency [4]. As an alternative, membrane separation technology looks to be more promising owing to the simplicity of operation, environmentally friendly nature, compact footprint, and energy efficient qualities [5]. Unfortunately, membrane technology, especially polymeric membranes, suffer a trade-off between permeability and selectivity [6]. In contrast,

inorganic membrane exhibits superior performance, but has challenges involved in the high cost and the complexity of fabrication. As a result, a composite membrane (CM) is developed by combining the benefits of the two types of membranes. This suggests that adding inorganic fillers to polymeric membranes will improve membrane performance due to the interaction of $CO_2$ gas with the fillers/polymers with the better compatibility between the polymer and filler.

Polymers derived from polar groups such as carbonyl, ether and acetate have a high affinity toward $CO_2$ gas [7]. Pebax-1657, for example, is a copolymer composed of solid polyamide and flexible polyether units in proportions of 60% and 40%, respectively [8]. Furthermore, the presence of polyether and polyamide compositions in the membrane will benefit $CO_2$ permeability and mechanical resistance [9]. In developing CM, inorganic fillers such as zeolites, carbon and metal-organic framework (MOF) fillers will be incorporated into the polymer matrix [10]. SAPO-34 fillers have been identified as a promising filler for the separation of gases with molecular sizes less than 0.38 nm, such as the separation of $CO_2$ gas (kinetic diameter ~0.33 nm), owing to the aperture diameter of SAPO-34 of 0.38 nm. In addition, SAPO-34 fillers also have high surface area, molecular sieve characteristics as well as excellent thermal and chemical stability [11]. Zhang et al. [12] fabricated the Pebax-1657/MFI CM for $CO_2$/$CH_4$ separation and reported a 63.5% and 76.4% increment in $CO_2$ permeability and $CO_2$/$CH_4$ selectivity as compared to the pristine Pebax-1657 membrane, owing to the great interfacial contact surface between polymer and filler. In another work, Si-CHA zeolite fillers were incorporated into the PEBAX-1657 polymer matrix by Ebadi et al. [13]. They discovered that the presence of Si-CHA zeolite fillers promotes $CO_2$ permeation better than $CH_4$ gas, resulting in the enhancement of $CO_2$ permeability and $CO_2$/$CH_4$ ideal selectivity ranging from 71 to 103%. Therefore, it is anticipated that the combination of Pebax-1657 and SAPO-34 for the formulation of CM would led to higher $CO_2$ separation from $C_2H_4$. Furthermore, with regard to date, no reported combination of Pebax-1657/SAPO-34 has been utilized for the ethylene recovery process.

Permeability and ideal selectivity are the important characteristics to assess the membrane performances. These two significant responses are varied depending on the operating conditions, including temperature and pressure, which affect the gas solubility and diffusivity [14–16]. Hence, the optimization of the operating conditions will lead to the enhanced solubility and diffusivity parameters, resulting in an optimum membrane performance. Furthermore, no associated studies on optimizing operating conditions for this new combination material, Pebax-1657/SAPO-34 CM, have been reported in the literature. Earlier research on composite membranes has concentrated on investigating the effect of different filler's concentration in the polymer matrix. In contrast, this work focuses on optimization of control conditions, including pressure and temperature, over the membrane gas performance, which has yet to be studied. Thus, by adopting a Pebax-1657/SAPO-34 CM, this study may be beneficial to the ethylene purification industries. The conservative methods used in optimization studies having some drawbacks including tedious and intensive-experimental works that are required for investigating operating variables [17]. Therefore, using an appropriate technique for optimization will minimize the need to repeat a similar process in order to achieve significant results. One reliable alternative is to utilize the RSM-CCD approach, which outperforms the conventional technique [18].

In this work, a new combination of Pebax-1657/SAPO-34 CM has been fabricated for $CO_2$/$C_2H_4$ separation. The morphology and elemental analysis of the resultant membrane were determined using FESEM and EDX. On the other hand, the relationship of operating conditions, including temperature and pressure, on gas permeability ($CO_2$ and $C_2H_4$) and $CO_2$ and $C_2H_4$ ideal selectivity was evaluated via the RSM-CCD approach using the DoE tool. Next, the numerical optimization of the operating conditions was validated by the experimental work. This work highlights on the performance optimization of a new membrane material combination composed of Pebax-1657 polymer matrix and SAPO-34 filler for ethylene recovery applications.

## 2. Materials and Methods

### 2.1. Materials

Both SAPO-34 particles and Pebax-1657 ((60 wt. % polyethylene oxide (PEO) and 40 wt. % polyamide (PA)) were bought from Arkema Inc. and Nova Scientific, respectively. Pure ethanol ($C_2H_6O$, Merck) and deionized water (Milli-Q) were used as solvent and non-solvent, respectively.

The pure $CO_2$ and $C_2H_4$ gases utilized in the gas permeation testing were supplied by Linde Malaysia Sdn Bhd.

### 2.2. Fabrication of CM

A 4 wt. % of Pebax-1657/SAPO-34 CM was fabricated via the solution casting method according to the reported literature [19]. SAPO-34 particles were first homogeneously mixed for 2 h at 90 °C under reflux in a solvent composed of 70% ethanol and 30% deionized water before being sonicated for 30 min. Following that, half of the total Pebax pellets were added to the SAPO-34 solution, which was then stirred for 24 h and sonicated for 30 min. The remaining polymer pellets were then added to the solution and vigorously stirred for 24 h before being sonicated for 30 min. The Pebax-1657/SAPO-34 solution was cast on a petri dish and dried for 24 h in a conventional oven at 50 °C. The membrane was then peeled off and dried for 24 h in a vacuum oven at 30 °C to remove any excess solvent.

### 2.3. Characterizations of CM

A 10 kV, a vacuum-operated field emission scanning electron microscope (FESEM, Zeiss Supra 55, Carl Zeiss NTS GmbH, Oberkochen, Germany) was utilized to examine the membrane's structure. Using a Q150R S model, platinum was sputtered and coated onto the membranes prior to imaging (Quasi-S Sdn. Bhd., Penang, MY). The presence of chemical elements in the resulting CM was examined using an Oxford Instrument Inca energy dispersion X-ray (EDX) equipped with FESEM (Oxford Instrument plc, Abingdon, UK).

### 2.4. Single Gas Permeation Measurements

The resultant CM was tested for gas separation performance by using gas permeation equipment. The permeability and selectivity of the membranes were measured at temperatures ranging from 25 °C to 60 °C and pressures of 3.5 to 10.0 bars, respectively. The permeation of $CO_2$ and $C_2H_4$ gases was determined using a gas permeation test apparatus as disclosed in the literature [19]. A bubble flow meter connected to a permeate line was used to check the flow rate of gas. The permeability, P of the membrane was determined by using Equation (1) as follows [20]:

$$P = \frac{V_P t}{A_m(p_h - p_l)} \tag{1}$$

where $V_p$ is the permeate flow rate ($cm^3$(STP)/s), $t$ is the membrane thickness (cm), $A_m$ is the membrane area ($cm^2$), $p_h$ and $p_l$ are the pressure in feed and permeate sides, respectively (cmHg). The permeability of the membrane is defined in unit Barrer (Barrer, 1 Barrer = $1 \times 10^{-10}$ $cm^3$ (STP).cm/s·$cm^2$·cmHg). The gas pair selectivity of CM was derived using Equation (2) as follows [21]:

$$\alpha_{CO_2/C_2H_4} = \frac{P_{CO_2}}{P_{C_2H_4}} \tag{2}$$

where $\alpha_{CO_2/C_2H_4}$ indicates the gas pair selectivity of the membrane and P stands for the permeability of the $CO_2$ and $C_2H_4$ gases (Barrer).

### 2.5. Optimization of CM Gas Separation Performance

The $CO_2/C_2H_4$ separation was optimized using DoE software, (Design-Expert V9.0, Stat-Ease Inc., Mpls). The CCD was chosen for its benefits such as flexibility, dependability,

and incessant operation. The current study used two operating conditions: temperature (A) and pressure (B), as well as three responses: permeability of gases ($CO_2$ and $C_2H_4$), and gas pair selectivity. All variables were investigated at three distinct measurement levels, from low to high. The alpha ($\alpha$) value was then set to 1, indicating face-centered design. The operating condition range and level are shown in Table 1. Referring to Table 1, the temperature (25 to 60 °C) and pressure (3.5 to 10.0 bar) ranges were chosen according to the conditions for ethylene recovery unit in the vinyl acetate monomer (VAM) production [22] are depicted. The DoE tool generated 13 runs of experiments from CCD, each with 4 factorial points, 4 axial points, and 5 center point repeats in a random order. For the lack of fit test, the duplicated center points are utilized to estimate pure error. The experimental run's varied sequence is designed to lessen the influence of uncontrolled circumstances. Therefore, 13 experiments have been performed, and three significant findings were obtained, including permeability of gases ($CO_2$ and $C_2H_4$) and gas pair selectivity, were measured.

**Table 1.** Experimental range and levels of the operational conditions.

| Operational Conditions | Units | Coded Measurement Levels | | |
|---|---|---|---|---|
| | | −1 (Low) | 0 (Center) | 1 (High) |
| Temperature (A) | °C | 25.0 | 42.5 | 60.0 |
| Feed Pressure (B) | Bar | 3.5 | 6.75 | 10.0 |

The correlation between operational factors such as temperature and pressure and responses such as gas permeability and $CO_2/C_2H_4$ ideal selectivity was further investigated using a statistical method known as analysis of variance (ANOVA). The experimental data were fitted into model parameters specified in Equation (3) to link their appropriate responses over operating situations [14].

$$Y = B_0 + \sum_{i=1}^{3} B_i x_i + \sum_{i=1}^{2} \sum_{j=i+1}^{3} B_{ij} x_i x_j + \sum_{i=1}^{3} B_{ii} x_i^2 \tag{3}$$

The statistically significant correlation of the operational parameters and interactions was evaluated using the fitted model's F-value and p-value for each response. The F-value is defined as the ratio of the mean square (MS) model and error. The MS model can be calculated by dividing the total squares by the df value. The p-value reflects the significance of the model, with a value of 0.05 indicating that the regression model is relevant and the null hypothesis (Ho): p 0.5 being rejected. Additionally, the consistency of the models was assessed using the coefficient of determination ($R_2$). The uncertainty that the fitted model cannot explain is known as the residual. In addition, the term "lack of fit" indicates that the lack of fit is negligible compared to pure error.

## 3. Results and Discussion

### 3.1. Physical Appearance of CM

Figure 1 demonstrates the physical appearance of the fabricated CM. It was observed that the fabricated CM was transparent and possessed good mechanical properties. The produced CM were free of cracks, resilient and flexible due to the higher composition of Pebax-1657 polymer than SAPO-34 zeolites in the formulation of CM. Soft PEO block in Pebax-1657 has a strong affinity for $CO_2$ and is very permeable. The membrane's structure and mechanical strength, on the other hand, are provided by a hard block of a semi-crystalline polymer. For CM, soft PEO blocks are also taken into account for the compatibility of the polymer matrix with an inorganic filler placed in it. This compatibility mainly results from PEO's strong chain mobility and the ether oxygen in its chemical nature.

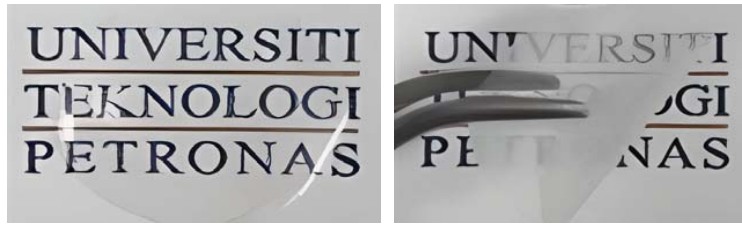

**Figure 1.** The physical appearance of the fabricated Pebax-1657/SAPO-34 CM.

*3.2. Characterizations of CM*

3.2.1. CM Morphology

The surface morphology of the resulting CM is shown in Figure 2. Referring to Figure 2, it was observed that the surface of the membrane is filled with considerably fewer aggregate clusters of SAPO-34 fillers. In addition, The FESEM image of the membranes also reveals that the membranes were successfully fabricated without significant interface defects such as voids, agglomeration, or non-uniformity of the membrane structure. Moreover, no observation of visible pores is detected in this image.

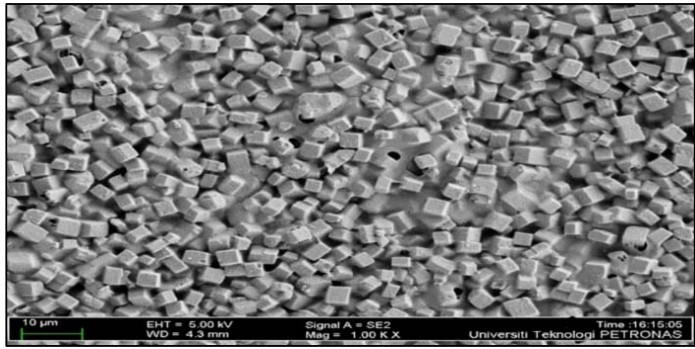

**Figure 2.** FESEM morphology of the membrane.

3.2.2. CM Elemental Mapping

Figure 3 shows the EDX data of Pebax-1657/SAPO-34 CM loaded with 4 wt.% of SAPO-34 fillers. Figure 3 shows that there was no particle aggregation and consistent dispersion of SAPO-34 particles within the Pebax-1657 polymer matrix. This demonstrates that SAPO-34 fillers successfully blended into the polymer matrix and had an excellent interface compatibility. The EDX results of the membrane, which showed the presence of aluminium (Al) and phosphorus (P) elements, further supported the conclusion that SAPO-34 fillers were present in the Pebax-1657 polymer matrix [23].

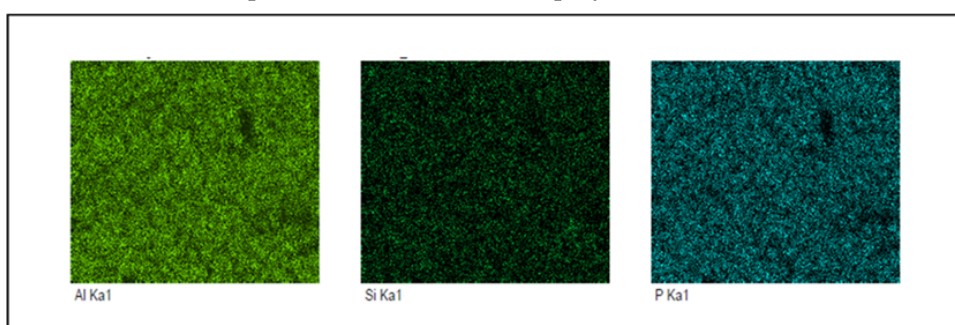

**Figure 3.** EDX analysis of fabricated membrane.

*3.3. Single Gas Permeation Performance*

Figure 4 shows the membrane separation performances of Pebax-1657/SAPO-34 CM at different feed temperatures. Referring to Figure 4, it can be observed that the permeability of $CO_2$ and $C_2H_4$ are in the range of 105.68 to 262.86 Barrer and 31.60 to 144.93 Barrer,

respectively. From Figure 4, the trends of $CO_2$ permeability are higher than $C_2H_4$ permeability. This could be due to the higher critical temperature of $CO_2$ (31.1 °C) than $C_2H_4$ (18 °C), indicating a higher rate of condensability of $CO_2$ gas, as well as $CO_2$ having additional interaction with polar groups in PEBAX-1657 polymer matrix [7]. On the other hand, lower $C_2H_4$ permeability associated to the bigger $C_2H_4$ kinetic diameter of 0.42 nm in comparison with $CO_2$ kinetic diameter of 0.33 nm. Apart from that, nonpolar $C_2H_4$ resulted in lower solubility in the composite membrane, thus leading to lower $C_2H_4$ permeability [24]. The addition of SAPO-34 filler increased the $CO_2/C_2H_4$ ideal selectivity from 2.13 to 3.52, resulting from the sieve effect role [25].

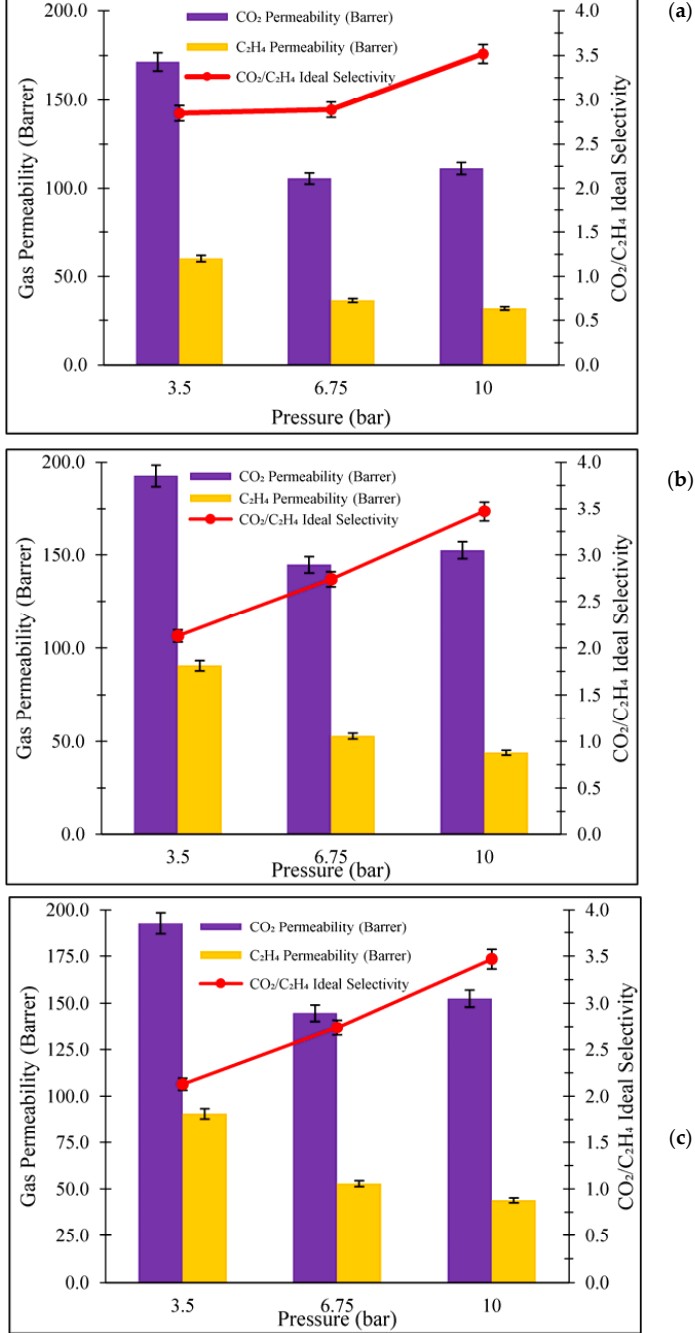

**Figure 4.** Gas separation performance of Pebax-1657/SAPO-34 CM at temperature of (**a**) 25 °C, (**b**) 42.5 °C and (**c**) 60 °C.

### 3.4. Central Composite Design (CCD)

Table 2 displays the condition of the operational parameters as well as the results of the experiments' significant reactions to the membrane separation performances. According to Table 2, the permeabilities of $CO_2$ and $C_2H_4$ are, respectively, between 105.68 and 262.86 Barrers and 31.60 and 144.93 Barrers.

**Table 2.** CCD and experimental findings.

| Path | A: Temperature (°C) | B: Pressure (bar) | R1: $CO_2$ Permeability (Barrer) | R2: $C_2H_4$ Permeability (Barrer) | R3: $CO_2/C_2H_4$ Selectivity |
|---|---|---|---|---|---|
| 1 | 25 | 10 | 111.34 | 31.6 | 3.52 |
| 3 | 25 | 6.75 | 105.68 | 36.62 | 2.89 |
| 9 | 25 | 3.5 | 171.5 | 60.21 | 2.85 |
| 2 | 42.5 | 6.75 | 155.92 | 52.13 | 2.99 |
| 4 | 42.5 | 6.75 | 155.92 | 52.13 | 2.99 |
| 5 | 42.5 | 6.75 | 144.77 | 42.86 | 2.74 |
| 6 | 42.5 | 6.75 | 144.77 | 42.86 | 2.74 |
| 11 | 42.5 | 10 | 152.59 | 43.96 | 3.47 |
| 12 | 42.5 | 3.5 | 192.72 | 90.66 | 2.13 |
| 13 | 42.5 | 6.75 | 144.77 | 52.86 | 2.74 |
| 7 | 60 | 10 | 238.74 | 63.27 | 3.26 |
| 8 | 60 | 6.75 | 201.89 | 93.32 | 2.16 |
| 10 | 60 | 3.5 | 262.86 | 144.93 | 1.81 |

3.4.1. Permeability of $CO_2$

The quadratic polynomial model for permeability of $CO_2$ given by the DoE tool is displayed in Equation (4) as a coded value.

$$CO_2 \text{ Permeability}_{coded} = 146.62 + 52.49 \text{ A} - 20.73 \text{ B} + 9.01 \text{ AB} + 13.68 \text{ A}^2 + 32.55 \text{ B}^2 \quad (4)$$

where A and B denote the temperature (°C), and pressure (bar), respectively.

Table 3 displays the ANOVA and regression analysis for $CO_2$ permeability over the membrane. Table 3 shows that the model's F and $p$-value, which, respectively, are 75.63 and 0.05, show that it is statistically significant. In this context, statistically significant model variables include A, B, $A^2$ and $B^2$. The $R_2$ value of 0.98 achieved, as shown in Table 3, attests to the suitability of the model for $CO_2$ permeability.

**Table 3.** ANOVA analysis for $CO_2$ permeability.

| Source | Sum of Squares | df | Mean Square | F-Value | *p*-Value |
|---|---|---|---|---|---|
| Model | 24562.06 | 5 | 4912.41 | 75.63 | 0.00001 [a] |
| A-Temperature | 16534.35 | 1 | 16,534.35 | 254.54 | 0.00000 |
| B-Pressure | 2579.64 | 1 | 2579.64 | 39.71 | 0.00040 |
| AB | 324.72 | 1 | 324.72 | 5.00 | 0.06045 |
| $A^2$ | 516.80 | 1 | 516.80 | 7.96 | 0.02575 |
| $B^2$ | 2926.09 | 1 | 2926.09 | 45.05 | 0.00027 |
| Residual | 454.70 | 7 | 64.96 | | |
| Lack of Fit | 305.51 | 3 | 101.84 | 2.73 | 0.17819 [b] |
| R-Squared | 0.98 | | | | |

[a] statistically significant, [b] statistically not significant.

Figure 5a displays a 3D plot of the effect of various operating conditions on $CO_2$ permeability. As seen in Figure 5a, increasing the temperature at a feed pressure of 3.5 bar resulted in increased $CO_2$ permeability of about 34.7% (from 171.5 Barrer to 262.86 Barrer). This might be owing to increased $CO_2$ diffusivity due to increased polymer chain mobility and free volume [26,27]. Meanwhile, increasing the pressure from 3.5 bar to 6.75 bar at

25 °C resulted in a reduction of $CO_2$ permeability (62.3%) due to the continual saturation of $CO_2$ gas inside polymer macro voids [28]. Furthermore, the $CO_2$ decrease may be attributed to lower fractional free volume (FFV) as a result of some reductions in transport routes [29]. However, the increment of feed pressure from 6.75 to 10.0 bar resulted in an enhancement of $CO_2$ permeability from 105.68 Barrer to 111.34 Barrer, indicating higher gas solubility [29]. The highest $CO_2$ permeability of 262.86 Barrer was reached at a temperature of 60 °C and a pressure of 3.5 bar. Meanwhile, the lowest $CO_2$ permeability of 105.68 Barrer was reached at 35 °C and 6.75 bar pressure. The parity plot for $CO_2$ permeability is depicted in Figure 4b, showing that the actual and anticipated response values are dotted in close proximity to the 95% prediction range.

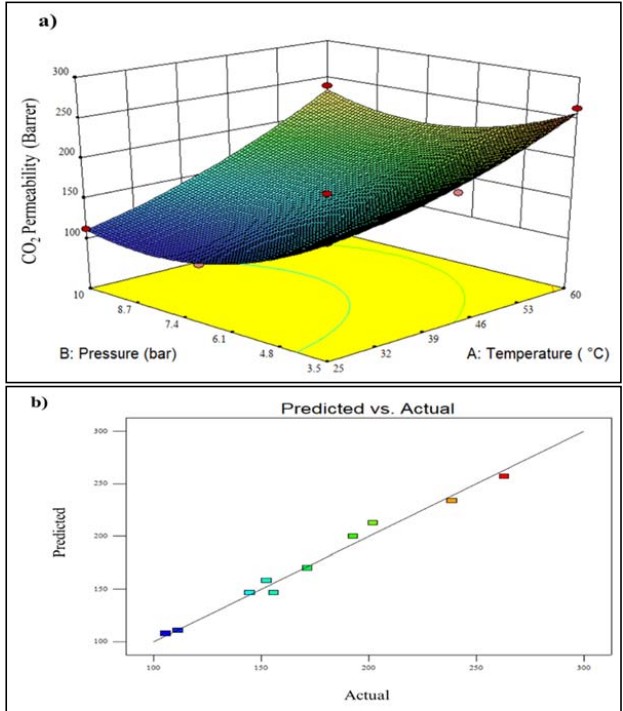

**Figure 5.** Plot of (**a**) 3D model of $CO_2$ permeability for temperature vs. pressure, and (**b**) predicted vs. actual $CO_2$ permeability (A lighter colour (blue) denotes lower membrane performance, whereas a darker colour (red) shows higher membrane performance.).

### 3.4.2. $C_2H_4$ Permeability

The quadratic polynomial model for $C_2H_4$ permeability given by the DoE tool is displayed in Equation (5) as a coded value.

$$C_2H_4 \text{ Permeability}_{coded} = 49.77 + 28.85\,A - 26.16\,B + 13.26\,AB + 12.20\,A^2 + 14.54\,B^2 \quad (5)$$

where A and B denote the temperature (°C) and pressure (bar), respectively.

Table 4 displays the ANOVA and regression analysis for $C_2H_4$ permeability over the membrane. Table 4 shows that the model's F and *p*-value, which, respectively, are 85.95 and <0.05, show that it is statistically significant. In this context, statistically significant model variables include A, B, AB, $A^2$ and $B^2$. The $R_2$ value of 0.98 achieved, as shown in Table 4, attests to the suitability of the model for $C_2H_4$ permeability.

Figure 6a shows a 3D plot of the effect of various operating conditions on $C_2H_4$ permeability. It can be seen that the permeability of $C_2H_4$ gas increases by 58.4% with increasing temperature, suggesting the enhancement of $C_2H_4$ diffusivity. This enhancement resulted from additionally accessible cavities and activation energy [26]. Furthermore, the permeability of $C_2H_4$ gas partly reduces from 60.21 Barrer to 31.6 Barrer as the feed pressure increases. The decrease in the permeability that occurs as feed pressure rise is most likely due to the membrane being more compacted under pressure, reducing the amount of free

volume in the membrane matrix [30]. Furthermore, for $C_2H_4$, the larger kinetic diameter of it makes it more difficult for $C_2H_4$ to diffuse through the membrane. Furthermore, the kinetic diameter of $C_2H_4$ (0.42 nm) gas is also larger than $CO_2$ gas (0.33 nm), leading to lower diffusivity compared to $CO_2$ gas. The lowest $C_2H_4$ permeability of 31.6 Barrer is observed at a temperature of 25 °C and a feed pressure of 10.0 bar. In contrast, the highest $C_2H_4$ permeability of 144.93 Barrer was obtained at 60 °C and 3.5 bar feed pressure (Figure 6b). The parity plot for $C_2H_4$ permeability is illustrated in Figure 6b, showing that the actual and anticipated response values are dotted in close proximity to the 95% prediction range.

**Table 4.** ANOVA analysis for $C_2H_4$ permeability.

| Source | Sum of Squares | df | Mean Square | F-Value | *p*-Value |
|--------|---------------|-----|-------------|---------|-----------|
| Model | 11,403.45 | 5 | 2280.69 | 85.95 | 0.000003 [a] |
| A-Temperature | 4993.36 | 1 | 4993.36 | 188.19 | 0.000002 |
| B-Pressure | 4106.60 | 1 | 4106.60 | 154.77 | 0.000004 |
| AB | 703.58 | 1 | 703.58 | 26.52 | 0.0013 |
| $A^2$ | 410.91 | 1 | 410.91 | 15.49 | 0.0056 |
| $B^2$ | 583.69 | 1 | 583.69 | 22.00 | 0.0022 |
| Residual | 185.73 | 7 | 26.53 | | |
| Lack of Fit | 76.78 | 3 | 25.59 | 0.94 | 0.5003 [b] |
| R-Squared | 0.98 | | | | |

[a] statistically significant, [b] statistically not significant.

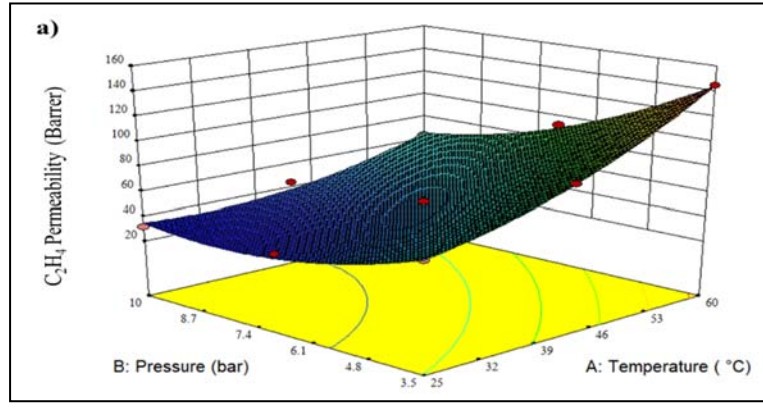

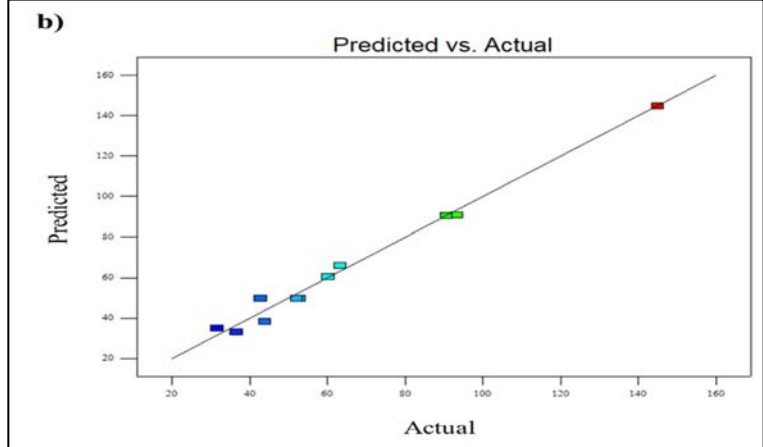

**Figure 6.** Plot of (**a**) 3D model of $C_2H_4$ permeability for temperature vs. pressure, and (**b**) predicted vs. actual $C_2H_4$ permeability (A lighter colour (blue) denotes lower membrane performance, whereas a darker colour (red) shows higher membrane performance.).

### 3.4.3. $CO_2/C_2H_4$ Ideal Selectivity

The quadratic polynomial model for $CO_2/C_2H_4$ ideal selectivity given by the DoE tool is displayed in Equation (6) as a coded value.

$$CO_2/C_2H_4 \text{Ideal Selectivity}_{coded} = 2.79 - 0.34\,A + 0.58\,B + 0.20\,AB - 0.13\,A^2 + 0.14\,B^2 \quad (6)$$

where A and B denote the temperature (°C), and pressure (bar), respectively.

Table 5 displays the ANOVA and regression analysis for $CO_2/C_2H_4$ ideal selectivity over the membrane. Table 5 shows that the model's F and $p$-value, which, respectively, are 20.34 and <0.05, show that it is statistically significant. In this context, statistically significant model variables include A and B. The $R_2$ value of 0.93 achieved, as shown in Table 5, attests to the suitability of the model for $CO_2/C_2H_4$ ideal selectivity.

**Table 5.** ANOVA analysis for $CO_2/C_2H_4$ ideal selectivity.

| Source | Sum of Squares | df | Mean Square | F-Value | *p*-Value |
|---|---|---|---|---|---|
| Model | 2.91 | 5 | 0.58 | 20.34 | 0.00049 [a] |
| A-Temperature | 0.69 | 1 | 0.69 | 24.01 | 0.00175 |
| B-Pressure | 2.00 | 1 | 2.00 | 69.75 | 0.00007 |
| AB | 0.15 | 1 | 0.15 | 5.32 | 0.05452 |
| $A^2$ | 0.05 | 1 | 0.05 | 1.73 | 0.22953 |
| $B^2$ | 0.05 | 1 | 0.05 | 1.92 | 0.20836 |
| Residual | 0.20 | 7 | 0.03 | | |
| Lack of Fit | 0.13 | 3 | 0.04 | 2.23 | 0.22744 [b] |
| R-Squared | 0.93 | | | | |

[a] statistically significant, [b] statistically not significant.

Figure 7a illustrates a 3D plot of the effect of various operating conditions on $CO_2/C_2H_4$ ideal selectivity. Referring to Figure 7a, it can be seen that the increment of temperature from 25.0 to 60.0 °C resulted in lower $CO_2/C_2H_4$ ideal selectivity. The reduction of $CO_2/C_2H_4$ ideal selectivity from 2.85 to 1.81 could be attributed to the higher chain mobility and free volume; thus, the permeation of larger molecules of gas ($C_2H_4$) is greater than smaller molecules ($CO_2$) [31]. In contrast, increasing the operating pressure from 3.5 to 10.0 bar at a temperature of 25 °C led to the improvement of $CO_2/C_2H_4$ ideal selectivity by about 19.0%. This could be due to the improvement of $CO_2$ adsorption capacity as well as better compatibility between polymer and filler [14]. Furthermore, the maximum $CO_2/C_2H_4$ ideal selectivity of 3.52 is attained at a temperature of 25.0 °C and a feed pressure of 10.0 bar, as shown in Figure 3a. In comparison, the lowest $CO_2/C_2H_4$ ideal selectivity of 1.81 was obtained at 60.0 °C feed temperature and 3.5 bar feed pressure. Figure 7b presents the parity plot for $CO_2/C_2H_4$ ideal selectivity, which shows that the actual and predicted response values are aligned within the 95% prediction range.

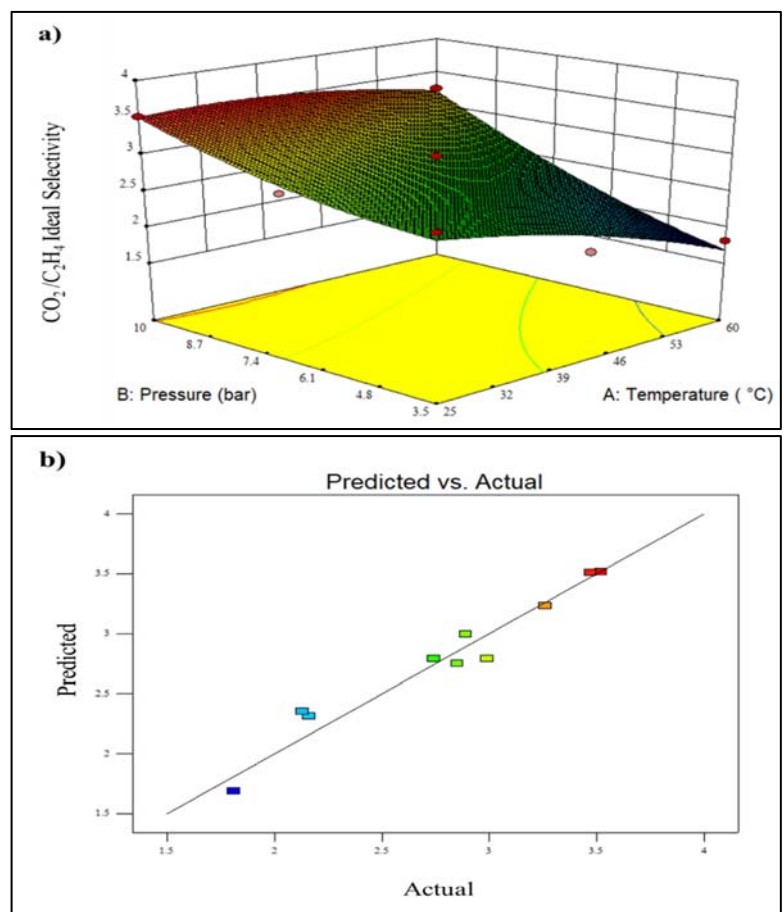

**Figure 7.** Plot of (**a**) 3D model of $CO_2/C_2H_4$ ideal selectivity for temperature vs. pressure, and (**b**) predicted vs. actual $CO_2/C_2H_4$ ideal selectivity (A lighter colour (blue) denotes lower membrane performance, whereas a darker colour (red) shows higher membrane performance.).

### 3.5. Numerical Optimization of $CO_2/C_2H_4$ Membrane Performances

The main goal of this research is to determine the optimal operating settings for improving the membrane separation performances. As a result, the optimization conditions were defined by maximizing $CO_2$ permeability and $CO_2/C_2H_4$ ideal selectivity for significant effects.

Table 6 displays the optimum approach and its desirability as determined by the DOE tool using actual data. The optimal temperature and pressure working settings were 60.0 °C and 10.0 bar, respectively, resulting in an optimum $CO_2$ permeability of 233.6 Barrer and $CO_2/C_2H_4$ ideal selectivity of 3.2.

**Table 6.** Numerical optimization via the RSM.

| Number | Temperature | Pressure | $CO_2$ Permeability (Barrer) | $C_2H_4$ Permeability (Barrer) | $CO_2/C_2H_4$ Selectivity | Desirability |
|---|---|---|---|---|---|---|
| 1 | 60.0 | 10.0 | 233.62 | 65.92 | 3.22 | 0.822 [a] |
| 2 | 59.5 | 10.0 | 231.03 | 64.77 | 3.24 | 0.817 |
| 3 | 60.0 | 9.5 | 226.50 | 67.76 | 3.07 | 0.755 |
| 4 | 52.5 | 10.0 | 198.11 | 51.05 | 3.28 | 0.735 |

[a] selected as optimum condition.

### 3.6. Validation of Numerical Optimization

To confirm the optimal, three further tests were run based on the suggested condition, and the results are shown in Table 7. From Table 7, in comparison to the standard deviation

of 1.1%, the $CO_2$ permeability error can be seen to range from 0.4% to 2.2%. The percentage error for the $CO_2/C_2H_4$ ideal selectivity, on the other hand, varies from 2.5% to 4.0%, with a standard deviation of 0.79%. For permeability of $CO_2$ and gas pair selectivity, the average errors were 1.67% and 3.10%, respectively.

**Table 7.** Experimental validation at feed pressure of 10 bar and temperature of 60 °C on thePebax-1657/SAPO-34 CM.

| Pressure (bar) | Temperature (°C) | $CO_2$ Permeability (Barrer) | | | $CO_2/C_2H_4$ Ideal Gas Selectivity | | |
|---|---|---|---|---|---|---|---|
| | | Experiment | Prediction | Error (%) | Experiment | Prediction | Error (%) |
| 10.0 | 60.0 | 238.68 | 233.62 | 2.20 | 3.31 | 3.22 | 2.80 |
| 10.0 | 60.0 | 230.42 | 233.62 | 1.40 | 3.35 | 3.22 | 4.00 |
| 10.0 | 60.0 | 234.50 | 233.62 | 0.40 | 3.30 | 3.22 | 2.50 |
| Mean (%) | | | | 1.67 | | | 3.10 |
| Standard Deviation | | | | 1.10 | | | 0.79 |

## 4. Conclusions

The current study centered on maximizing working parameters such as the temperature and pressure of the constructed membrane, Pebax-1657/SAPO-34 CM. The physical appearance of the fabricated CM was transparent and possessed good mechanical properties owing to the hard polyamide phase, which provides mechanical strength, while the soft polyether phase provides the transport channel for the filler to permeate through the pores of the Pebax-1657 polymer matrix, which made it hard to break. Furthermore, FESEM and EDX images of the CM provided evidence that the inorganic SAPO-34 fillers were well distributed in the Pebax-1657 polymer matrix, and good dispersion of the particles between the polymeric and inorganic phases was observed. Also, elements of SAPO-34 which consist of aluminium, silicon and phosphorus, were present in the resulting CM, confirming the presence of SAPO-34 fillers in a polymer matrix. The gas separation performance of the resultant CM was measured, and it was discovered that the permeability of $CO_2$ and $C_2H_4$ gases is governed by the temperature, but for $CO_2/C_2H_4$, the ideal selectivity is impacted by the pressure. From RSM analysis, the optimal operating conditions achieved a maximum $CO_2$ permeability of 238.68 Barrer and a $CO_2/C_2H_4$ ideal selectivity of 3.31 at a temperature of 60.0 °C and a feed pressure of 10.0 bar. The average errors for the $CO_2/C_2H_4$ ideal selectivity and $CO_2$ permeability were 3.10% and 1.67%, respectively, indicating a 95% reliability for the model. The resulting $R_2$ values were in the 0.93 to 0.98 range, demonstrating the statistical significance of the regression models. Overall, the combination of Pebax-1657 and SAPO-34 CM in this work demonstrated a hopeful potential for the utilization in $CO_2$ and $C_2H_4$ gas separation applications. Nonetheless, further parametric analysis is needed before considering a scale-up study.

**Author Contributions:** Conceptualization, N.H.S. and N.J.; methodology, N.H.S. and N.J.; software, N.J.; validation, N.H.S. and S.S.R.; formal analysis, N.H.S. and S.S.R.; investigation, N.H.S. and S.S.R.; resources, N.J.; data curation, N.H.S.; writing—original draft preparation N.H.S. and S.S.R.; writing—review and editing, N.H.S., C.W.M.C. and N.J.; visualization, N.H.S.; supervision, N.J. and N.S.S.; project administration, N.J.; funding acquisition, N.J. All authors have read and agreed to the published version of the manuscript.

**Funding:** The financial support provided by JRP RESEARCH GRANT (Cost center: 015MDO-080) and Yayasan Universiti Teknologi PETRONAS (YUTP) RESEARCH GRANT (Cost center: 015LC0-270) are duly acknowledged.

**Institutional Review Board Statement:** Not applicable.

**Data Availability Statement:** Not applicable.

**Acknowledgments:** The technical support provided by $CO_2$ Research Centre ($CO_2$RES), Institute of Contaminant Management is duly acknowledged.

**Conflicts of Interest:** The authors declare no conflict of interest.

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
