# Peer review of "Ethylene Recovery via Pebax-Based Composite Membrane: Numerical Optimization"

_sustainability, doi:10.3390/su15031856_

Round 1

Reviewer 1 Report

Comments and Suggestions for Authors

The manuscript entitled “Ethylene Recovery via Pebax-based Composite Membrane: Numerical Optimization” deals with the recovery of ethylene via Pebax-based Composite Membrane. The manuscript was submitted by Nadia Hartini Suhaimi, Norwahyu Jusoh, Syafeeqa Syaza Rashidi, Christine Wei Mann Chng and Nonni Soraya Sambudi can be considered and will be accepted for publication in “Sustainability” Journal after a minor revision. The manuscript is well written with all essential information.The results are worth to be published but needs some improvement.

Following are the comments and suggestions which need to be addressed.

1.    The application of the present research should be highlighted in the manuscript and explain the real-time application of Pebax-based composite membrane concerning a large scale.

2.    The rationale why the Pebax-based composite membrane was selected for this study?

3.    In the introduction, part research gap and the clear objective of the present study are missing. The author should be added in the introduction part.

4.    Please carefully revise the manuscript to remove the grammatical errors (line no. 70-72) and vague sentences. There are some misspellings and improper writing styles in the context that should be revised. English needs to be carefully checked again and polished.

5.    The author should add the comparisons of specific experimental data obtained with the results of the authors of other works in the results and discussion section. An author should add a high-resolution graphical image.

6.    To increase the validity of your represented data you need to repeat them with at least 3 replications and then add error bars to your data point in almost all presented graphs in this study.

7.    The author should add the cost analysis for this present research work.

8.    The author should add the BET surface area, particle size, pole volume, EDAX, XRD, and TGA analysis of the composite membrane.

9.    How to reuse the membrane? The author should incorporate the reusability of the composite membrane.

10.    The author could explain the yield and rejection rate of the membranes in the results and section. Give a mechanism to recover ethylene using a Pebax-based Composite Membrane.

I believe that the manuscript can be accepted for publication with minor corrections.

Author Response

Dear Reviewer,

Kindly refer to the attached file.

Thank You.

Reviewer 2 Report

The authors focused on a research topic with high applicability and practical importance. The efficiency of gas mixture separation processes is a widely studied issue, and attempts to optimize the gases separation by using mathematical models are particularly useful approaches. The research is properly conceptualized, the aim of the work is well defined and the novelty is pointed out. Methodology is structured in distinct subchapters and is presented accurately. The results are presented and analyzed with professionalism and are accompanied by eloquent images. The authors studied and cited very recent scientific articles. Conclusions are drawn up based on the main experimental results, while the need for further studies is emphasized as well.

Author Response

(The authors gave the same response as above.)

Reviewer 3 Report

The present work addresses the development of a composite membrane comprising of Pebax-1657 polymer matrix and SAPO-34 fillers and tests its performance for CO2/C2H4 separation. From my point of view, the paper presents an adequate theoretical framework and is well-discussed; however, I think the manuscript must be carefully checked for grammar and spelling.

I highlight below some sentences that need to be revised:

Line 18: “…..ranging from 25.0 - 60.0 °C and 3.5 - 10.0 bar……”

Lines 71-72: “…the pristine Pebax-1657 membrane, owing to the bigger (bigger?) interfacial contact surface between polymer andfiller.”

Lines 73-74: “They discovered that the presence of Si-CHA zeolite fillers promotes CO2 permeation that CH4 gas,…”

Line 147: “…the permeability of CO2 and CH4 gases (Barrer)”. Was it methane or ethylene?

Lines 187-188: “The produced CM were free of cracks, resilient and flexible, .due to the higher composition of Pebax-1657 polymer l than SAPO-34 zeolites, in the formulation of CM.”

Lines 213-214: “Figure 3 shows the EDX data of Pebax-1657/SAPO-34 CM loaded with 4 wt. %. With the elements of aluminium (Al), silicon (Si) and phosphorus (P).”

Line 342: “…the percentage inaccuracy for the CO2/C2H4 ideal selectivity, , ranges….”

And so on...

Author Response

(The authors gave the same response as above.)

Reviewer 4 Report

1- The significance of the present work should be highlighted in the last paragraph of the introduction.

2- The authors are required to highlight the gap of the work especially the RSM method within the manuscript.

3- How did the authors select the RSM parameters and their minimum and maximum values? Please explain it. 

4- It is strongly recommended to compare the outcome of the present work with the up to date literature.

Author Response

(The authors gave the same response as above.)
